# COVID-19 cases among medical laboratory services staff in South Africa, 2020–2021: A cohort study

**Kerry Sidwell Wilson**[1,2]*, **Vusi Ntlebi**[1], **Felix Made**[1], **Natasha Sanabria**[3], **Melissa Vetten**[3], **Jitcy Joseph**[3], **Graham Chin**[4], **David Jones**[4], **Nonhlanhla Tlotleng**[1]

**1** Epidemiology and Surveillance Section, National Institute for Occupational Health, A Division of National Health Laboratory Service, Braamfontein, Johannesburg, South Africa, **2** Faculty of Health Sciences, School of Public Health, University of the Witwatersrand, Witwatersrand, South Africa, **3** Toxicology and Biochemistry Department, National Institute for Occupational Health, A Division of National Health Laboratory Service, Braamfontein, South Africa, **4** National Safety Health and Environment Department, National Institute for Occupational Health, A Division of National Health Laboratory Services, Braamfontein, South Africa

* KerryW@nioh.ac.za

**Data Availability Statement:** De-identified participant data containing possibly identifiable data and personal medical data may be obtained on application to the National Health Laboratory

## Abstract

Medical laboratory workers may have an increased risk of COVID-19 due to their interaction with biological samples received for testing and contamination of documents. Records of COVID-19 laboratory-confirmed positive cases within the medical laboratory service were routinely collected in the company's Occupational Health and Safety Information System (OHASIS). Surveillance data from the OHASIS system were extracted from 1 April 2020 to 31 March 2021. An epidemic curve was plotted and compared to that for the country, along with prevalence proportions and incidence rates. The odds of COVID-19 infection were categorised by job and compared to the US Occupational Risk Scores. A logistic regression model assessed the risk of COVID-19 infection per occupational group. A total of 2091 (26% of staff) COVID-19 positive cases were reported. The number of COVID-19 cases was higher in the first wave at 46% (967/2091) of cases, than in the second wave 40% (846/2091) of cases. There was no significant difference in COVID-19 prevalence between male and female employees. The job categories with the most increased risk were laboratory managers [AOR 3.2 (95%CI 1.9–5.1)] and laboratory support clerks [AOR 3.2 (95%CI 1.9–5.2)]. Our study confirms that some categories of medical laboratory staff are at increased risk for COVID-19; this is a complex interaction between workplace risk factors, community interaction, socioeconomic status, personal habits, and behaviour. Targeted interventions are recommended for high-risk groups. OHASIS has the potential to generate data for surveillance of health care workers and contribute towards a South African risk profile.

Services and with compliance to their requirements and the POPI Act South Africa 2021. The application may be made on registering at https://aarms.nhls.ac.za/NHLS_AARMS/Public/Default.aspx. Please contact Aarms.Support@nhls.ac.za with any queries. The data set is the OHASIS incident dataset 2020/21 with key variables "outcome_COVID", Occupation, and demographic variables.

**Funding:** The authors received no specific funding for this work.

**Competing interests:** I have read the journal's policy and the authors of this manuscript have the following competing interests: One author is involved in establishing the information system used here to provide the data in the local government health departments. This does not alter our adherence to PLOS ONE policies on sharing data and materials.

## Introduction

The coronavirus disease 2019 (COVID-19) pandemic has affected many organisations across the world, including healthcare systems (hospitals, care homes, and medical laboratories), with increased workloads and risk of infection [1]. Healthcare workers (HCWs), including laboratory staff, serve on the front lines during outbreaks to detect, control, and slow down the transmission of the disease [2,3].

Due to the nature of their work, HCWs are at an increased risk of contracting Severe acute respiratory syndrome coronavirus (SARS-COV-2), as was observed in Wuhan, China, where the daily rate was three times higher in HCWs [2]. The South African Department of Employment and Labour has classified HCWs and laboratory personnel responsible for collecting or handling COVID-19 specimens as having high exposure risk [4]. Laboratory support staff, such as messengers and drivers, and cleaners, may also be at increased risk of COVID-19 as they may handle patient specimens and contaminated outer packaging, as well as documents, or be exposed to other staff through shared workspaces and facilities [2,5]. Staff sharing facilities may also be at increased risk [6]. However, the South African guideline for essential services stated that equipment and stationery might not be shared, and all goods should be disinfected before use [7]. A possible route of occupational exposure for COVID-19 in laboratory workers includes aerosol-generating procedures (AEP), surface contamination on the primary specimen containers and specimen carrier bags, as well as environmental contamination in the laboratories where samples are collected [5].

The Occupational Health and Safety Information System (OHASIS) is used to record and investigate health and safety incidents of the South African medical laboratory services staff. Traditionally, HCWs have been classified as nurses, doctors, paramedics, porters, allied health, and Laboratory scientists. However, medical laboratory services staff may be a more inclusive term to represent all those involved in the various duties reported on OHASIS and the study herein. Due to the high risk of contracting COVID-19, OHASIS has been used to monitor all incidents occurring in the workplace, including the incidence of COVID-19 infections amongst employees [8]. Both workplace and community-acquired infections were recorded. OHASIS is a comprehensive online occupational health and safety information programme that can control access to information of a confidential nature, such as medical, personal, and health and safety information. It has the potential to generate data for surveillance in South Africa and contribute towards a draft of a South African risk profile.

Several reports are available on COVID-19 in HCWs in South Africa, particularly those based in hospitals [9,10]. Nonetheless, there is a lack of reports focusing on the subcategories of health workers, such as medical laboratory workers, who may have a different education around exposure and type of interaction with the disease. This study, therefore, aimed to describe the distribution of COVID-19 cases among medical laboratory staff using the OHASIS system as a surveillance reporting tool. The medical laboratory is a South African national government institution, which provides various laboratory and related public health services for 80% of the population through a network of laboratories in the country.

## Methods

### Study design

A COVID-19 investigation based on routine event-based surveillance and cohort analysis involving the staff within a large national medical laboratory service within South Africa for the duration of the first 12 months (i.e., 1 April 2020–31 March 2021) of the COVID-19 pandemic in the country.

## Population

All NHLS staff, including contract workers and experiential students, may report incidents to the surveillance system and, thus, be included in this study. Managers, safety health and environment staff and occupational health nurses may also report incidents reported to them on the surveillance system. Thus, all staff were eligible to be included in this analysis.

## Inclusion criteria

- Reported case on OHASIS.

- Employed, working or visiting at one of the medical laboratory facilities.

## Exclusion criteria

- Age less than 18 years.

## Surveillance

The OHASIS system was introduced in 2011 in a paper format, after which it was upgraded to an online system. OHASIS records all workplace incidents (near misses, accidents, exposures, and disease) in staff, contractors, and visitors. OHASIS also captures demographic information, continuously updated from the human resources database. Reports of injuries, near hits, and occupational diseases are received on paper for capture or directly entered into the system by a user or self-reported. The surveillance information system is coupled with strategic education and teaching programmes to enhance occupational health and safety.

To reduce underreporting during COVID-19, telephonic reporting to the occupational health and safety staff was added to OHASIS. Existing modules were amended, and a new submodule was added in May 2020 to OHASIS to deal with COVID-19 reporting and allow staff to screen daily for symptoms of COVID-19. The symptom screening checklist includes reporting of body pain/aches, fever/chills, including the option of recording actual temperature measurement, cough, shortness of breath/chest discomfort, sore throat, loss of smell or taste, redness of eyes, nausea, vomiting, diarrhoea, and unexplained fatigue, weakness or tiredness. A positive symptom screen will lead to a consultation with a medical professional, and if deemed appropriate, the employee will be referred for a COVID-19 test. Any staff member testing positive for COVID-19 must provide a list of all co-workers they interacted with, considering exposure risk factors to affect contact tracing.

All high-risk contacts were required to quarantine for a minimum of 7 days. During this time, the employee self-monitored for COVID-19 symptoms and, if they remained asymptomatic, underwent an RT-PCR COVID-19 test to determine an early return to work, in line with the national guidance for health workers. A positive test resulted in the employee going into isolation for the mandatory ten days. A high-risk contact was anyone less than one meter apart from the person who tested positive for more than 15 minutes without a mask. Low-risk contacts would return to work but continued monitoring for symptoms and applying stringent non-pharmaceutical COVID-19 interventions.

## Safety protocols for laboratory staff

In addition to the mandatory screening mentioned above, numerous other interventions against COVID-19 were implemented. These included all staff being supplied with two cloth

masks, in addition to two surgical masks per day, with appropriate biohazard waste disposal posted at entrances/exit points for discarding used surgical masks. Hand sanitiser stations were placed at crucial identified locations. Transparent barriers to separate shared workspaces were erected. Signage and floor markings to promote social distancing were placed in commonly used areas, including tea rooms. Shift rotation was implemented to facilitate social distancing, and the number of staff entering the workplace was aligned to the declaration of the national lockdown level at the particular time.

## Data collection

COVID-19 RT-PCR test positive data and demographic data for this study were extracted from the OHASIS database for the period in question. Staff records, including age, sex, job title, and location, were extracted from the human resources department as denominator data in the calculations. Where feasible, monthly staff data was used. The job categories listed in OHASIS for medical laboratory services staff in South Africa are shown in Tables 1 and 2. Ethical clearance was obtained for OHASIS data access from the University of the Witwatersrand human research ethics committee. A secondary analysis was conducted using this data, which had been anonymised and grouped to comply with the Protection of Personal Information Act South Africa. De-identified participant data containing possibly identifiable data and personal medical data may be obtained on application to the National Health Laboratory Services and with compliance to their requirements and the POPI Act South Africa 2021. The application may be made on registering at https://aarms.nhls.ac.za/NHLS_AARMS/Public/Default.aspx.

## The COVID-19 risk score

This score is based on US data only. It is different from the Occupational Risk Score, which is calculated by the USA Labour Department itself. The score for each occupation was calculated to evaluate the data on three job requirements covered in the occupational database. Firstly, contact with others was taken into account, e.g., how much does this job require the worker to be in contact with others to perform it? Secondly, physical proximity was considered, e.g., to what extent does this job require the worker to perform tasks in close physical proximity to others? Lastly, the exposure to disease and infection was determined, e.g., how often does this job require exposure to hazardous infectious conditions? Each attribute was assigned a weight, then aggregated to arrive at a final COVID-19 Risk Score between 0 and 100, with 100 representing the highest possible risk. Jobs with a risk score below 0.5 were excluded from further analysis. Occupations that were held by fewer than 20,000 people were removed. From the remaining pool, 100 well-known occupations were selected. The assessment identified the most at-risk workers, descending from highest to lowest risk (www.onetonline.org/). The score was represented as a range (Table 2), which spans the lowest and highest score, for all possible jobs listed by the United States Department of Labour (DOL), within that broad job description. This USA data range was then compared to the job categories listed in OHASIS (Tables 1–3).

## Statistical analysis

Statistical analysis was carried out in the STATA version 16 statistical package (StataCorp, USA). Weekly incidence rates of laboratory-confirmed cases were presented for the South African general population, Gauteng and Western Cape Provinces and medical laboratory services staff as reported in OHASIS (i.e., national medical laboratory service within South Africa staff). The number and the proportion of cases were presented for each month, from April

**Table 1. Description of the medical laboratory staff included in this analysis 2020–21.**

| Risk Factors | N | No. Cases (proportion per staff employed) | OR (95%CI) | AOR (95% CI) |
|---|---|---|---|---|
| **Sex** | | | | |
| Male | 2863 | 652 (22.8) | 0.96 (0.87–1.07) | 0.93 (0.83–1.03) |
| Female | 6109 | 1438 (23.5) | ref | |
| **Race** | | | | |
| African | 6781 | 1584(23.4) | 2.4 (1.88–3.02) | 2.6 (2.0–3.3) |
| Chinese | 9 | 0 (0) | - | - |
| Coloured | 647 | 151 (23.3) | 2.3 (1.72–3.09) | 2.0 (1.48–2.73) |
| Indian | 520 | 82 (15.8) | 1.4 (1.03–1.20) | 1.4 (0.99–1.97) |
| Caucasian | 715 | 80 (11.2) | ref | ref |
| missing | 301 | 194 (64.5) | | |
| **Age** | | | | |
| <29 | 1830 | 355 (19.4) | ref | ref |
| 30–39 | 3378 | 824 (24.4) | 1.3 (1.14–1.51) | 1.5 (1.29–1.74) |
| 40–49 | 2080 | 507 (24.4) | 1.3 (1.1–1.52) | 1.6 (1.34–1.85) |
| 50–59 | 1217 | 278 (22.8) | 1.2 (1.01–1.44) | 1.5 (1.27–1.85) |
| 60+ | 391 | 82 (21.0) | 1.1 (0.83–1.42) | 1.4 (1.01–1.87) |
| **Province** | | | | |
| Eastern Cape | 865 | 326 (37.7) | 2.5 (2.13–2.92) | 2.5 (2.13–2.97) |
| Free State | 466 | 102 | 1.3 (1.01–1.62) | 1.4 (1.12–1.18) |
| Gauteng | 3732 | 656 | ref | ref |
| KwaZulu Natal | 1811 | 501 | 1.7 (1.47–1.91) | 1.8 (1.53–2.03) |
| Limpopo | 426 | 87 | 1.2 (0.95–1.56) | 1.2 (0.99–2.29) |
| Mpumalanga | 228 | 71 | 1.9 (1.46–2.59) | 1.8 (1.34–2.35) |
| North West | 266 | 74 | 1.8 (1.34–2.35) | 1.7 (1.31–2.46) |
| Northern Cape | 140 | 33 | 1.4 (0.95–1.56) | 1.5 (0.99–2.29) |
| Western Cape | 1039 | 241 | 1.4 (1.17–1.63) | 1.8 (1.48–2.18) |

2020 to March 2021. Statistical tests of proportion were carried out to compare the proportions of cases by occupation for the medical laboratory services staff.

An epidemic wave was defined to include an upward and downward period for a sustained time in a country. The waves of COVID-19 infection in the South African public data were identified by the National Institute for Communicable Disease (NICD) when the weekly national numbers of new infections rose above 30 cases per 100 000. The wave ended when the infections dropped below that number [11]. This definition was not used in this data due to the much higher rate of cases. Thus, a wave was identified when infections rose above 300 per

**Table 2. Infection rate of COVID-19 reported in OHASIS for medical laboratory staff, by occupation group over 12 months.**

| Occupation Group | No of staff | Cases | Infection rate over 12 months |
|---|---|---|---|
| Administrative and Clerical | 952 | 207 | 21,7% |
| General Worker | 1054 | 237 | 22,5% |
| Laboratory Manager or supervisor | 493 | 140 | 28,4% |
| Laboratory Staff—Skilled | 3537 | 790 | 22,3% |
| Laboratory Staff—Unskilled | 1801 | 539 | 29,9% |
| Medical Staff | 1138 | 178 | 15,6% |

**Table 3. Prevalence proportion of COVID -19 cases within the medical laboratory services staff and within international HCWs job categories.**

| Job Category | Positive Cases | Prevalence (%) | Adjusted Odds Ratio (95%CI) | COVID-19 Occupational risk score (range)** |
|---|---|---|---|---|
| **Administrative and Clerical** | 146 | 23.4 (146/622) | **2.3 (1.4–3.9)** | 22.0–42.6 |
| **General Worker** | 64 | 17.1 (64/375) * | 1.5 (0.9–2.6) | **74.9** (Groundsman and maintenance have no matches) |
| **Cleaner** | 49 | 24.3 (29/170) | **2.2 (1.3–4.0)** | 22.7–40.4 |
| **Messenger** | 124 | 26.0 (124/477) | **2.5 (1.5–4.1)** | (no matches found) |
| **Management** | 32 | 20.0 (32/160) | **2.2 (1.2–4.0)** | (no matches found) |
| **IT Technology Personnel** | 29 | 17.1 (29/170) | 1.5 (0.8–2.9) | 4.7–23.43 |
| **Laboratory Manager or Supervisor** | 140 | 28.4 (140/493) * | **3.2 (1.9–5.2)** | 23.8–33.4 |
| **Medical Technician** | 270 | 29.2 (270/925) * | **2.9 (1.8–4.7)** | **70.7–84.1** |
| **Medical Technologist** | 428 | 24.1 (429/1783) | **2.5 (1.5–4.0)** | **62.5–80.6** |
| **Medical Scientist** | 40 | 11.3 (40/354) * | 1.1 (0.6–1.9) | (no matches found) |
| **Other Laboratory Unskilled staff** | 45 | 23.8 (45/189) | **2.1 (1.2–3.8)** | (no matches found) |
| **Laboratory Intern** | 51 | 10.7 (51/475) * | 0.1 (0.03–0.3) | (no matches found) |
| **Laboratory Clerks** | 494 | 30.6 (494/1612) * | **3.2 (1.9–5.1)** | (no matches found) |
| **Medical Officers** | 9 | 16.1 (9/56) | 0.5 (0.1–1.9) | 23.6 |
| **Nurse** | 16 | 24.6 (33/267) | 1.7 (0.7–3.7) | **60.9–86.1** |
| **Registrar** | 33 | 12.4 (33/267) * | 1.4 (0.7–2.5) | 23.6 |
| **Pathologists** | 22 | 9.1 (22/243) | Reference group | 23.6 |
| **Phlebotomists** | 98 | 19.3 (22/507) * | 1.6 (1.0–2.8) | **68.2** |

*proportion test compared to the mean rate 23.3% (* is significant at p< 0.05%); Adjusted OR (AOR) was adjusted for age, race, province and sex.

**ONET, where the range spans the lowest and highest score, for all possible jobs listed by the US DoL, within that broad job description (www.visualcapitalist.com/the-front-line-visualizing-the-occupations-with-the-highest-covid-19-risk/).

100 000 in daily cases. An unconditional logistic regression was conducted to investigate the risk of COVID-19 infection within each occupation category within the medical laboratory services staff adjusting for risk factors such as age, sex, ethnicity, and province.

## Limitations

The information provided by OHASIS is limited by the quality of the data entry and the information provided by the staff. There is possible reporting bias in the data, particularly around possible community exposure, as well as in cases for contracted staff or students, where they may feel vulnerable and less secure than permanent staff when describing situations and responsibility. The investigations conducted by trained nurses, who were employed to provide occupational health services, mitigate some of these issues along with the requirement for managers to report all cases to the nurses for investigation. Variations in infection proportions by province that do not always correspond the proportion seen in the general population, have suggested variations in reporting procedures and, perhaps, access to testing for the general public.

## Results

The study cohort consisted of an average of 8121 medical laboratory staff employed across South Africa for the 12 months of this study. The staff included in this study either conducted laboratory tests or provided support for the laboratories. By 31 March 2021, a total of 2091 laboratory-confirmed cases of COVID-19 were reported, providing an overall infection rate of 25.7%.

The medical laboratory service employs mainly females (68.1%), but no significant relationship was seen between sex and COVID-19 cases in the staff (AOR 0.93 95%CI 0.83–1.03). All race groups were represented, where African workers represented 75.5% of the staff. Compared to Caucasian workers, African and Coloured workers had significantly increased odds of COVID-19 infection. The majority of workers were between 30 and 49 years of age (60.8%) and all age groups compared to those less than 29 years showed significantly increased odds of infection. Odds of infection varied significantly between some provinces, with the Eastern Cape reporting the highest odds compared to Gauteng province. (Table 1).

As of 30 April 2021, South Africa experienced two waves of COVID-19. The first wave was identified from 7 June 2020 to 22 August 2020 and peaked in weeks 28–29 of 2020 (11 July–18 July 2020). The second wave was from the 15 November 2020 to the 6 February 2021 and peaked in week 1 of 2021 (26 December 2020–2 January 2021). Two primary waves were identified in the OHASIS weekly data (Fig 1), corresponding to the South African general public. However, the daily incidence rate for medical laboratory staff increased to above 30 per 100000 staff in epidemic week 2020.17 (the week of 20 April 2020). It did not drop below this threshold until epidemic week 2021.12 (21 March 2021). The number of HCW COVID-19 cases in the first wave was 967 (46% of total cases) and in the second wave 846 (40% of total cases). There were no significant differences in the sex and age of the staff who were infected in the first and second waves (p = 0.944 and p = 0.2673, respectively).

Due to the higher rate of COVID-19 cases in the medical laboratory staff data, waves were identified using a daily case rate of 300 per 100 000 staff (S1 Fig). Two earlier waves of

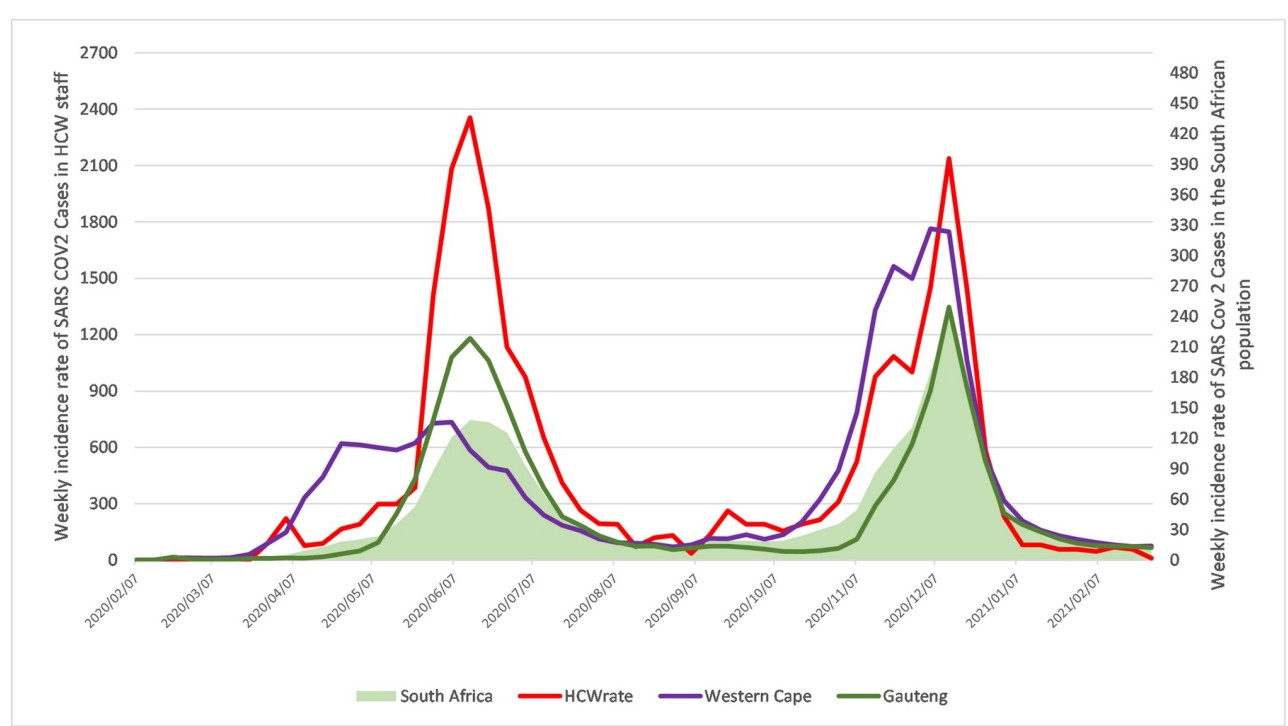

**Fig 1. An epidemic curve of the incidence of COVID-19 cases within medical laboratory services staff compared to the general South African population.** Weekly incidence rates of PCR confirmed cases of SARS CoV-2 by epidemic week for the South African population (green shading, right axis), medical laboratory services staff (red line, left axis), Gauteng province population (green line, right axis), and the Western Cape province population (blue line, right axis). 1 April 2020–31 March 2021.

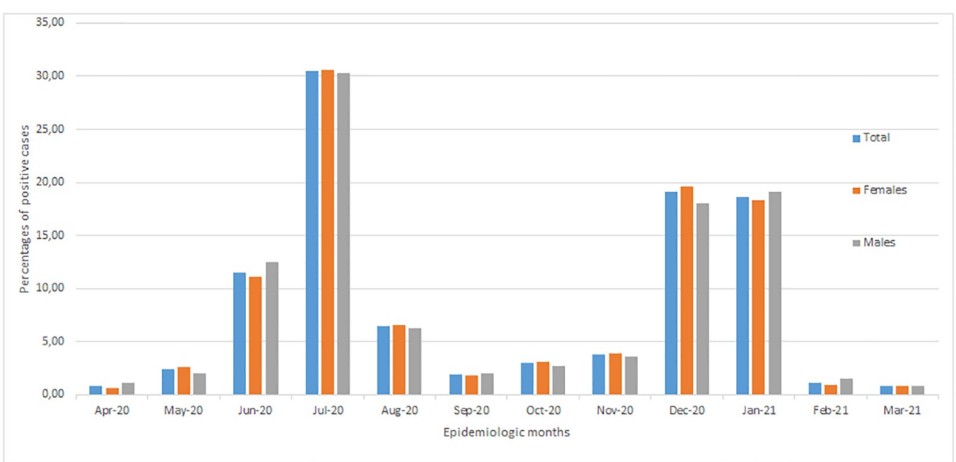

**Fig 2. Monthly number and proportion of infections and incidence rates among the medical laboratory services staff by sex, 1 April 2020–31 March 2021.**

COVID-19 infections were experienced during May 2020 (i.e., before the main wave, which began on the 4 June 2020) one week before the country entered the first wave of COVID-19, where the medical laboratory services staff also peaked one week before the rest of the country (week of the 7 July 2020). This first wave did not end until 11 September 2020, well after the country left the first wave. Again, the medical laboratory services staff entered the 2nd COVID -19 wave early on 13 October 2020, one month before the country entered the second wave.

The distribution of the proportion of positive COVID-19 cases across months in the last calendar year and by sex are presented in Fig 2. The highest proportion of positive cases was recorded in July 2020, followed by December 2020 and January 2021. The highest rate of positive COVID-19 cases was observed in July 2020. While April 2020, February 2021 and March 2021 recorded the lowest proportion of positive cases. No difference in the incidence rates between men and women was observed.

Fig 3 shows both the number of COVID-19 positive cases reported by the province, as well as the corresponding percentage of COVID-19 positive cases amongst staff within the laboratories. Interestingly although Gauteng province reported the highest number of positive cases (586 cases), this province showed the lowest proportion of infected staff (16%).

Occupation groups provide information on the type of work, socioeconomic status and indirectly on the education of the staff. Health workers, including laboratory workers, messengers, and phlebotomists, would be more likely to be exposed to COVID -19 through their interaction with biological samples. However, all medical laboratory staff may be exposed to COVID-19 through interaction with colleagues, both in and outside the workplace. Although the "COVID-19 Risk Score" that was based on US data excluded jobs with a risk score below 0.5 from further analysis (Table 3), none of the occupations listed in South African OHASIS dataset were excluded from analysis, where occupations with few employees were grouped into similar occupation groups (Table 2). The occupational group referred to as "unskilled laboratory staff" showed the highest rate of COVID-19 (29.9%) followed by laboratory managers (28.4%), while medical staff had the lowest rate at 15.6% (Table 2).

Occupation groups were subdivided further based on job category (Table 3), where the highest rates of COVID-19 infection were seen in laboratory clerks (30.6%), medical technicians (29.2%) and laboratory managers or supervisors (28.4%). In contrast, the medical staff,

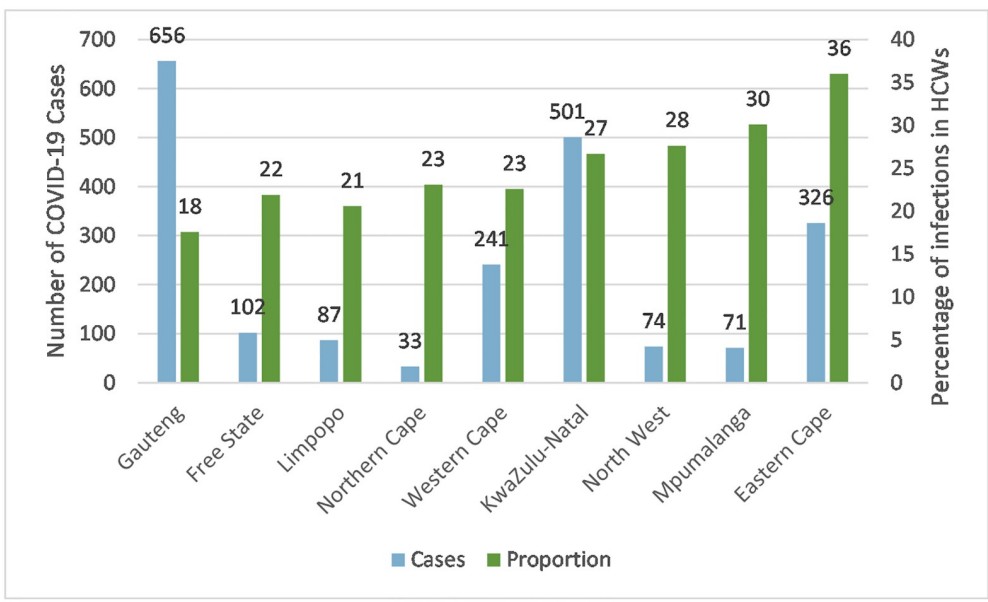

**Fig 3. Proportion of COVID-19 positive cases among medical laboratory services staff by province, 1 April 2020–31 March 2021. (chi-square *p*< *0.001*).**

medical scientists, general workers, and pathologists suffered significantly lower rates. Similarly, American pathologists, registrars and medical officers also had a lower COVID-19 occupational risk score. As pathologists reported the lowest proportion of infections, they were chosen as the reference group for this study.

After adjusting for age, sex, province of work and race, laboratory managers and laboratory clerks were found to have over three times the odds compared to pathologists, with an AOR 3.2 95%CI (1.9–5.20) AOR 3.2 95%CI (1.9–5.1), respectively. Also increased were medical technicians AOR 2.9 95%CI (1.8–4.7). Medical technologists and messengers were both 2.5 times at risk [AOR 2.5 95%CI (1.5–4.0)] and [AOR 2.5 95%CI (1.5–4.1)] respectively. Staff with over twice the risk were found to include the management staff [AOR 2.2 95%CI (1.2–4.0)], administrative and clerical workers [AOR 2.3 95%CI (1.4–3.9)], as well as the other unskilled laboratory staff [AOR 2.1 95%CI (1.2–3.8)]. The cleaners had the lowest risk [AOR 1.5 95%CI (0.9–2.6)] of COVID-19 infection.

## Discussion

This study provided information on the transmission rate of infections in an under-researched group of HCWs, medical laboratory services staff, in South Africa, via an Occupational Health and Safety Information System (OHASIS). The study presented herein assessed the epidemiological distribution of laboratory-confirmed SARS-CoV-2 cases from 1 April 2020 to 31 March 2021, detected in the staff of a large medical laboratory in South Africa, during the first two waves of the COVID-19 pandemic (Figs 1–3). This study also reported the prevalence and infection rates of COVID-19 in different occupational groups within these laboratories (Tables 1, 2 and 3). This study has value because various studies have outlined an excessive transmission risk of SARS-CoV-2 in HCWs [12–14], but few have specifically assessed prevalence and infection rate in medical laboratory services staff. A substantial proportion (65%) of medical laboratory staff work in the laboratories, while the remainder provide support services, e.g.,

HR, procurement, and IT services. Thus, individuals in these laboratory-based occupations may be exposed to COVID-19 (via the patient samples received for testing and through transmission from colleagues) if proper safety protocols (e.g., social distancing and PPE) were not maintained during work or community exposures.

As of 31 March 2021, a total of 2091 laboratory-confirmed COVID-19 positive cases were reported on OHASIS, which represented 26% of the laboratory workers. This proportion is much higher than the 2.6% prevalence reported for the general South African population for the same period [15]. A higher proportion of HCW cases is not unexpected, as the study by Gomez-Ochoa and colleagues showed a higher prevalence of SARS-CoV-2 infection among HCWs compared with the data from the general population [16]. Similar to reports on HCWs in South Africa, where more cases and increased COVID-19 hospital mortality were seen [9,17]. Compared to the epidemic curve among the population in SA, the epidemic curve comprising COVID-19 cases reported on OHASIS showed two waves (Fig 1). The number of COVID-19 cases in the first wave was slightly higher (46% of total cases) than cases reported in the second wave (40% of total cases). A similar decrease in COVID-19 cases among medical laboratory services staff was seen in the second wave. This was thought to be due to reasons including improved safety protocols being implemented in health facilities, availability and appropriate use of PPE, better infection control measures being put into place, as well as improved handling of admitted patients [9]. In addition, previous exposure to COVID-19 has been shown to result in subsequently improved immunity against infections, which may also be the reason for a reduced number of cases in the second wave [18].

The South African Government enacted a national lockdown effective on 27 March 2020. The National lockdown activities included social distancing, a complete shutdown of childcare, primary and higher education institutions, enabling employees to work from home, reduction of shopping facilities, and limited transport services [19]. However, medical laboratory services staff were considered essential and at work during this national lockdown to process and analyse specimens of undiagnosed, asymptomatic patients or positive COVID-19 patients. This could be the reason for the increased number of COVD-19 cases among medical laboratory staff in April and May 2020 compared to the SA population before the first wave (Fig 1). It is feasible that the medical laboratory workers were exposed at the beginning of the COVID-19 pandemic due to handling COVID-19 patient samples and exposure to colleagues [20] because the community-based infection rates increased only after that and to a lesser degree. However, many workers may have become infected outside of the workplace during the growing transmission in the community [14]. It should also be noted that the group of HCW medical laboratory staff in this study had access to more RT-PCR testing for COVID-19 (under a robust occupational health program) than compared to the general South African population, which implies a testing bias and may explain the observed findings.

A significant difference in the incidence rates between men and women was not found (Fig 2). This data also compared age groups, where significant differences were observed with those 30 to 60 years at more risk than those under 30 years. Previous studies have alluded to a significant role of sex and age in COVID-19 related mortality [9]. In another study, it was found that men who worked in healthcare occupations in England and Wales had a statistically higher rate of death involving COVID-19 (44.9 deaths per 100,000 males; 190 deaths) when compared with the rate of COVID-19 among men of the same age in the population [21]. The rate among women who worked in healthcare occupations (17.3 deaths per 100,000 females; 224 deaths) was statistically similar to the rate in the population [21].

After adjusting for age, sex, province of work, and race, it was found that irrespective of the high-, intermediate-, and low-exposure risk settings of the HCW per region, all were at a significantly increased risk of COVID-19 infection compared to pathologists. All occupation

groups, medical technicians, medical technologists, laboratory managers, administrative and clerical workers, cleaners, messengers, unskilled laboratory staff, laboratory clerks, and management suffered a higher risk. This finding is similar to that of Folgueira et al., who reported no significant differences in the infection rates between the different groups of HCWs working in high-, intermediate-, and low-exposure risk settings [22].

Many laboratory workers, messengers, and phlebotomists would be exposed to COVID-19 through their work with biological samples, while all staff may be exposed to COVID-19 through exposure to colleagues at work. Although some infections may arise at work, others may be acquired through contact with an infected person in the household or communities, where safe social distancing and safety measures were not maintained during non-working days. In Ontario, Canada, the rate of COVID-19 in HCWs was 5.5 times higher than the general population but was found to be in parallel with the infections in the general population [23]. They found HCWs were also more likely to present asymptomatically or with atypical symptoms, where the authors suggested a testing bias and underestimation of the burden in the general population. Gomez-Ochoa and colleagues reported that only a few studies analysed the potential source of infection, limiting the possibility of evaluating the impact of nosocomial versus community-acquired infection [16]. They further reported that Hunter et al. found no differences in the proportion of infected HCWs when comparing the ones with patient-facing roles with those without this exposure [24]. In contrast, Zheng and colleagues found that clinical staff had a higher infection rate than non-clinical staff [25]. However, these authors did admit that they found a decrease in the rates after staff were advised on how to use PPE and the authors state that PPE was effective in reducing transmissions when used appropriately. A study conducted in a Turkish hospital found significant risk factors for infection were the presence of a SARS-CoV-2 positive person in the household (P = 0.0160), inappropriate use of PPE while caring for patients (P = 0.003), and failure to keep a safe social distance from HCW (P = 0.003). However, other specific risk factors related to habits were mentioned, such as occupying the same personnel break room as HCWs without a medical mask for more than 15 min (P = 0.000) or consuming food within 1 metre of an HCW (P = 0.0030) [26]. The studies of García-Basteiroet et al., Kluytmans et al., and Sikkema et al. provided evidence suggesting a relevant role of community transmission of disease in HCW infections [27–29]. Similarly, a study investigating the characteristics of COVID-19 in HCWs in a London hospital found that the numbers mostly followed the community [26]. In other words, household contacts may play a significant role in SARS-CoV-2 infection in HCWs, mainly due to the rapid circulation of the virus in the community. This idea was investigated by Shah and colleagues, where patient-facing HCW were at higher risk and the household members of patient-facing healthcare workers [30]. They further determined that the risk of admission due to COVID-19 in non-patient-facing healthcare workers (and their households) was similar to the risk in the general population. The OHASIS data reported for the medical laboratory services staff per region (Fig 3) highlight that this is a complex interaction between workplace risk factors, community interaction, socioeconomic status, personal habits and behaviour.

Most countries, including South Africa, were unprepared for handling new infections; hence a rapid increase in cases was observed for HCWs in the first wave [31,32]. There was also a global shortage of PPE in the early months of COVID-19 and inequitable access between countries to PPE [33]. However, many other factors are related to the elevated risk of COVID-19 infection in HCWs, including excessive workload, lack of infection control measures, and inappropriate use of PPE [31,32,34]. All South African HCWs monitored by OHASIS, especially medical laboratory services staff, were mandated to wear sufficient PPE when contacting patients with confirmed or expected COVID-19. All the staff were trained in infection control

practices, and the use of PPE was provided by the internal (Safety, Health and Environment) SHE department, which managed OHASIS implementation.

Despite the precautionary measures implemented, it can be seen in Table 2 that a total of 2091 staff had confirmed COVID-19 cases during the period of study. The infection rate was 0.8% in April 2020, and this climbed gradually until the highest infection rate was recorded in July 2020 (30.5%); thereafter, the rate decreased moderately. However, this number peaked again in the second wave, where the infection rate was 19.9% in December 2020. Factors that put laboratory workers at risk must be identified and mitigated to improve health and safety, job satisfaction, and productivity [2]. The International Federation of Clinical Chemistry and Laboratory Medicine (IFCC) convened a task force to guide laboratory practitioners. A simulation study was performed to test the impact of recommended prevention methods in laboratory workers [35]. The simulation recommended over and above standard recommendations, fewer staff per shift with others working from home, frequent staff changes or mutually exclusive working teams and fewer consecutive workdays, as well as the use of N95 respirators, gloves, and gowns where possible. In the US, occupations with the highest COVID-19 risk were classified as general practitioners, registered nurses and radiologic technicians, among others, but they also categorised pathologists as at lower risk (www.onetonline.org/).

Since pathologists reported the lowest proportion of infections, they were chosen as the reference group for this study. In Table 2, the American pathologists, registrars and medical officers had a lower COVID-19 occupational risk score, while medical technicians, medical technologists, general workers and nurses recorded a higher COVID-19 risk score. Among South African medical laboratory workers, the highest rates of COVID-19 infection were seen in laboratory clerks (30.6%), medical technicians (29.2%) and laboratory managers or supervisors (28.4%). It is clear from the OHASIS data that the laboratory managers had an unexpectedly high proportion of infection. The increase in managers might be due to the personal habits and behaviour, or work requirements where greater interaction occurred between managers and their staff, a high volume of clinical handovers took place, or they dealt with contaminated paper-based work. Laboratory personnel handle paperwork daily from the various hospitals, often within a short time from when the form was completed. The virus can be viable for more than 24 hrs at room temperature, and it is not easy to sanitise the paper considering the content. Therefore, alternate novel interventions to those discussed above from similar studies should be implemented. For example, a study investigating the characteristics of COVID-19 in HCWs in a London hospital found that novel interventions for replacing paper-based specimen receiving with electronic-based work and the use of ultraviolet irradiation to sterilise the papers received in the laboratory may help reduce the risk of infection [25,36,37]. These are recommendations that could be implemented within the South African context.

Initially, two distinct pre-waves in COVID-19 rates were observed before the reported first wave event (S1 Fig). The OHASIS data indicated that the HCW first wave started one week before the general public South African population first wave. This observation prompted the idea that the OHASIS monitoring system may be a plausible early warning mechanism. This held true for the second wave, where the OHASIS data showed the wave started in the HCW staff one month before the country's second wave. Nonetheless, there is a lack of reports focusing on the subcategories of health workers, such as medical laboratory workers, who may have a different education around exposure and type of interaction with the disease. In addition, there is a lack in reporting where only a few studies analysed the potential source of infection, limiting the possibility of evaluating the impact of nosocomial versus community-acquired infection [16]. Hence, research gaps have been identified where improved monitoring and surveillance tools are needed within medical settings to act as early warning systems. As an early

warning system, OHASIS data is based on investigations conducted for incidents and then recorded, allowing for data reliability and monitoring and interventions in real time.

## Conclusion

HCWs, specifically medical laboratory services staff, have been crucial to monitoring COVID-19 infections and other diseases. The ongoing protection of the health of this group of workers is vital for the continued effectiveness of the healthcare system. This study suggested that with increased access to testing, the actual population prevalence of COVID-19 in South Africa could be as high as 25%. Furthermore, the study confirmed that some medical laboratory workers are at an increased risk of COVID-19, requiring further investigation. Targeted interventions for high-risk occupations were recommended. The implementation of a robust screening and reporting system for workers for COVID-19, as described herein, should be continued as an effective monitoring and surveillance tool, where systems similar to OHASIS may even act as an early warning system.

## Supporting information

**S1 Fig. Daily case rate and 7day moving average rate for the medical laboratory services staff.** The daily case rate showed a large fluctuation over weekends where testing was seldom available (green) the 7-day moving average showed two clear peaks during the period (blue) the orange line represents the 30/100 000 cut off for a wave.
(TIF)

## Acknowledgments

The National Health Laboratory Service (NHLS) SHE Department for their hard work investigating and monitoring all staff during the pandemic.

## Author Contributions

**Conceptualization:** Kerry Sidwell Wilson, David Jones.

**Data curation:** Kerry Sidwell Wilson.

**Formal analysis:** Kerry Sidwell Wilson, Felix Made, Nonhlanhla Tlotleng.

**Investigation:** Kerry Sidwell Wilson, Natasha Sanabria, Melissa Vetten, Nonhlanhla Tlotleng.

**Methodology:** Kerry Sidwell Wilson, Felix Made, Graham Chin, Nonhlanhla Tlotleng.

**Project administration:** Kerry Sidwell Wilson.

**Supervision:** Graham Chin, David Jones.

**Visualization:** Kerry Sidwell Wilson, Nonhlanhla Tlotleng.

**Writing – original draft:** Kerry Sidwell Wilson, Natasha Sanabria, Melissa Vetten, Jitcy Joseph.

**Writing – review & editing:** Kerry Sidwell Wilson, Vusi Ntlebi, Felix Made, Natasha Sanabria, Melissa Vetten, Jitcy Joseph, Nonhlanhla Tlotleng.

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
