## [Decision Letter · Decision Letter 0]

31 Jan 2022

PONE-D-21-29566COVID-19 cases among medical laboratory services staff in South Africa, 2020-2021: A Cohort Study.PLOS ONE

Dear Dr. Kerry Sidwell Wilson,

Thank you for submitting your manuscript to PLOS ONE. After careful consideration, we feel that it has merit but does not fully meet PLOS ONE’s publication criteria as it currently stands. Therefore, we invite you to submit a revised version of the manuscript that addresses the points raised during the review process.

We look forward to receiving your revised manuscript.

Kind regards,

Massimiliano Galdiero, M.D., Ph.D.

Academic Editor

PLOS ONE

Journal Requirements:

I have read the journal's policy and the authors of this manuscript have the following competing interests: One author is involved in establishing the information system used here to provide the data in the local government health departments.   

Reviewers' comments:

Reviewer's Responses to Questions

**Comments to the Author**

1. Is the manuscript technically sound, and do the data support the conclusions?

Reviewer #1: Yes

Reviewer #2: Yes

2. Has the statistical analysis been performed appropriately and rigorously? 

Reviewer #1: Yes

Reviewer #2: Yes

3. Have the authors made all data underlying the findings in their manuscript fully available?

Reviewer #1: Yes

Reviewer #2: Yes

4. Is the manuscript presented in an intelligible fashion and written in standard English?

Reviewer #1: Yes

Reviewer #2: No

5. Review Comments to the Author

Reviewer #1: COVID-19 cases among medical laboratory services staff in South Africa, 2020-2021: A Cohort Study.

Technical Comments to the Author

The manuscript is written in clear and understandable English and the final message is very strong.

Remarks to the Author

I suggest minor comments.

Minor comments

1. Remove point from the title. The title should be “COVID-19 cases among medical laboratory services staff in South Africa, 2020-2021: A Cohort Study”

2. Report in the materials and methods section a paragraph on the inclusion and exclusion criteria used in the study.

3. Move limitations to the Materials and Methods section.

4. A better description of the study population is needed. Include Supplementary Table 1 as Table 1 in the manuscript and comment on it in the results section.

5. Indicate occupations with risk scores below 0.5 excluded from the analysis.

6. The description of the results needs to be improved. For example, figure 2 shows the incidence rates of infection among medical laboratory service personnel by gender, not all data are commented on in the text.

7. The captions of the figures appear within the text. They should be placed at the end of the paragraph in which the figure is cited.

8. Improve the resolution of the figures.

9. Change HCWS to HCWs in conclusion part (line 460).

10. HCWs has been critical for monitoring COVID-19 infections and other diseases, not for diagnosis. Please correct (lines 460-461).

Reviewer #2: Summary: The aim of this study was to show an increased risk of COVID-19 for Medical laboratory workers. Records of COVID-19 laboratory-confirmed positive, in twelve months, were present in System (OHASIS). The results showed that some categories of medical laboratory staff are at increased risk for COVID-19, with the most increased risk were laboratory managers, confirms this is a complex interaction between workplace risk factors. This article is very interesting for the public of this journal. The language of paper is difficult to understand and includes many errors, and the results must be better reported. The main text and the data analysis have to need to improve.

1) In Abstract, result section: “… A total of 2091 COVID-19 positive cases (26% of staff) were reported via OHASIS over the twelve months’ period. Already present in materials and methods.

i.e.,967 (46% of total cases), write in full i.e., write in full ei, such as reported is little clary. ...The number of COVID-19 cases was higher in first wave, i.e.,967 (46% of total cases) than in the second wave 846 (40% of total cases)...this part is very confusional; improve, for example putting the fraction in the parenthesis, and number percentual out of them, n% (n°/n. tot).

2) Line 133: we are missing parentheses and not in Grasset character.

3) Improve the introduction reporting similar studies.

4) Line 293: delete r, before the point..

5) Line 368: 1 min, no 1m.

6) Line 429, 460: correct it .

7) Improve the figures, and in my opinion it would be better to remove the totals bar.

6. PLOS authors have the option to publish the peer review history of their article (what does this mean?). If published, this will include your full peer review and any attached files.

Reviewer #1: No

Reviewer #2: No

---

## [Author Response · Author response to Decision Letter 0]

24 Feb 2022

Revision Letter 

Thank you for allowing us to revise the manuscript and resubmit. We confirm that we have addressed each of the Reviewer’s comments (please see details below). We have also ensured that the manuscript meets the PLOS ONE style requirements. In addition, we have updated the Methods section to include the Data Availability statement. The ethics statement has also been moved to the Methods section, as requested. We have provided the images, as requested. 

Journal Requirements

1. Style – The submission style was updated to match the formatting guidelines. 

2. Competing interests: One author is involved in establishing the information system used here to provide the data in the local government health departments. This does not alter our adherence to PLOS ONE policies on sharing data and materials. Deidentified participant data containing possibly identifiable data and personal medical data may be obtained on application to the National Health Laboratory Services and with compliance to their requirements and the POPI Act South Africa 2021. The application may be made on registering at https://aarms.nhls.ac.za/NHLS_AARMS/Public/Default.aspx

3. Data availability - Deidentified participant data may be obtained from the National Health Laboratory Services on application and compliance with their requirements. https://aarms.nhls.ac.za/NHLS_AARMS/Public/Default.aspx

4. Your ethics statement should only appear in the Methods section of your manuscript. If your ethics statement is written in any section besides the Methods, please delete it from any other section. – ethics statement removed from Acknowledgements and stated in Methods section

Reviewers Comments 

Reviewer #1: COVID-19 cases among medical laboratory services staff in South Africa, 2020-2021: A Cohort Study.

Technical Comments to the Author

The manuscript is written in clear and understandable English and the final message is very strong.

Thank you for this view

Remarks to the Author

Minor comments

1. Remove point from the title. The title should be “COVID-19 cases among medical laboratory services staff in South Africa, 2020-2021: A Cohort Study”

Point removed, title now reads “COVID-19 cases among medical laboratory services staff in South Africa, 2020-2021: A cohort study”

2. Report in the materials and methods section a paragraph on the inclusion and exclusion criteria used in the study.

The paragraph below on population was included in the Methods section, i.e., 

“Population

All NHLS staff, including contract workers and experiential students, may report incidents to the surveillance system and, thus, be included in this study. Managers, safety health and environment staff and occupational health nurses may also report incidents reported to them on the surveillance system. Thus, all staff were eligible to be included in this analysis.

Inclusion criteria: 

• Reported case on OHASIS

• Employed, working or visiting at one of the medical laboratory facilities 

Exclusion criteria

Age less than 18 years”

3. Move limitations to the Materials and Methods section.

Limitations paragraph moved from the Discussion section to the Methods section 

A better description of the study population is needed. Include Supplementary Table 1 as Table 1 in the manuscript and comment on it in the results section.

Table S1 moved to results section and renamed Table 1.

We then included in the results section “The medical laboratory service employs mainly females (68.1%), but no significant relationship was seen between sex and COVID-19 cases in the staff (AOR 0.93 95% CI 0.83 - 1.03). All race groups were represented, where African workers represented 75.5% of the staff. Compared to Caucasian workers, African and Coloured workers had significantly increased odds of COVID-19 infection. The majority of workers were between 30 and 49 years of age (60.8%) and all age groups compared to those less than 29 years showed significantly increased odds of infection. Odds of infection varied significantly between some provinces, with the Eastern Cape reporting the highest odds compared to Gauteng province. (Table 1)”

All following table numbers were also updated. 

4. Indicate occupations with risk scores below 0.5 excluded from the analysis.

Although the “COVID-19 Risk Score” that was based on US data excluded jobs with a risk score below 0.5 from further analysis (Table 3), none of the occupations listed in South African OHASIS dataset were excluded from analysis, where occupations with few employees were grouped into similar occupation groups (Table 2). This statement has been added to the main text.

6. The description of the results needs to be improved. For example, figure 2 shows the incidence rates of infection among medical laboratory service personnel by gender, not all data are commented on in the text.

No difference in the incidence rates between men and women was seen. This statement has been added to the main text.

7. The captions of the figures appear within the text. They should be placed at the end of the paragraph in which the figure is cited. 

The caption for Fig 1 moved to below the paragraph it is first mentioned in. The remaining two figure captions are correctly placed. 

8. Improve the resolution of the figures.

Thank you for noticing. The resolution has been improved to 300dpi

9. Change HCWS to HCWs in conclusion part (line 460). Thank you for catching this typo. It has been corrected to HCWs.

10. HCWs has been critical for monitoring COVID-19 infections and other diseases, not for diagnosis. Please correct (lines 460-461). Diagnosing removed from the sentence.

Reviewer #2: Summary: The aim of this study was to show an increased risk of COVID-19 for Medical laboratory workers. Records of COVID-19 laboratory-confirmed positive, in twelve months, were present in System (OHASIS). The results showed that some categories of medical laboratory staff are at increased risk for COVID-19, with the most increased risk were laboratory managers, confirms this is a complex interaction between workplace risk factors. This article is very interesting for the public of this journal. The language of paper is difficult to understand and includes many errors, and the results must be better reported. The main text and the data analysis have to need to improve.

Thank you for your comments. We have completed a spell check to correct spelling errors and have revised grammatical mistakes where needed. The results section has been revised.

1) In Abstract, result section: “… A total of 2091 COVID-19 positive cases (26% of staff) were reported via OHASIS over the twelve months’ period. Already present in materials and methods.

Sentence changed to: A total of 2091 (26% of staff) COVID-19 positive cases were reported.

i.e.,967 (46% of total cases), write in full i.e., write in full ei, such as reported is little clary. ...The number of COVID-19 cases was higher in first wave, i.e.,967 (46% of total cases) than in the second wave 846 (40% of total cases)...this part is very confusional; improve, for example putting the fraction in the parenthesis, and number percentual out of them, n% (n°/n. tot).

Thank you for your suggestion. The sentence was corrected to “The number of COVID-19 cases was higher in the first wave at 46%(967/2091) of cases, than in the second wave 40% (846/2091) of cases”.

2) Line 133: we are missing parentheses and not in Grasset character.

Parentheses included, I am unfortunately not familiar with Grasset characters. 

3) Improve the introduction reporting similar studies.

Thank you for this suggestion. Currently this is the only report of COVID-19 incidence in South Africa focusing on the subcategories of health workers, such as medical laboratory workers. 

4) Line 293: delete r, before the point..

Thank you for your suggestion. The sentence was deleted

5) Line 368: 1 min, no 1m.

 The abbreviation was corrected to 1 metre

6) Line 429, 460: correct it .

Line 429 corrected to “These are recommendations that could be implemented within the South African context.”

Line 460 corrected to “HCWs, specifically medical laboratory services staff, have been crucial to monitoring COVID-19 infections and other diseases”

7) Improve the figures, and in my opinion it would be better to remove the totals bar.

Thank you for your comment. The totals bar was removed from Fig 2 and the resolution improved to 300dpi in all figures.

---

## [Decision Letter · Decision Letter 1]

13 May 2022

COVID-19 cases among medical laboratory services staff in South Africa, 2020-2021: A Cohort Study

PONE-D-21-29566R1

Dear Dr. Sidwell Wilson,

We’re pleased to inform you that your manuscript has been judged scientifically suitable for publication and will be formally accepted for publication once it meets all outstanding technical requirements.

Kind regards,

Massimiliano Galdiero, M.D., Ph.D.

Academic Editor

PLOS ONE

Additional Editor Comments (optional):

Reviewers' comments:

Reviewer's Responses to Questions

**Comments to the Author**

1. If the authors have adequately addressed your comments raised in a previous round of review and you feel that this manuscript is now acceptable for publication, you may indicate that here to bypass the “Comments to the Author” section, enter your conflict of interest statement in the “Confidential to Editor” section, and submit your "Accept" recommendation.

Reviewer #1: All comments have been addressed

Reviewer #2: All comments have been addressed

2. Is the manuscript technically sound, and do the data support the conclusions?

Reviewer #1: Yes

Reviewer #2: Yes

3. Has the statistical analysis been performed appropriately and rigorously? 

Reviewer #1: Yes

Reviewer #2: Yes

4. Have the authors made all data underlying the findings in their manuscript fully available?

Reviewer #1: Yes

Reviewer #2: Yes

5. Is the manuscript presented in an intelligible fashion and written in standard English?

Reviewer #1: Yes

Reviewer #2: Yes

6. Review Comments to the Author

Reviewer #1: The authors have adequately addressed the comments raised in the review process and I believe this manuscript is improved and acceptable for publication.

Reviewer #2: (No Response)

7. PLOS authors have the option to publish the peer review history of their article (what does this mean?). If published, this will include your full peer review and any attached files.

Reviewer #1: No

Reviewer #2: No

---

## [Editor Report · Acceptance letter]

10 Jun 2022

PONE-D-21-29566R1 

COVID-19 cases among medical laboratory services staff in South Africa, 2020-2021: A cohort study 

Dear Dr. Wilson:

I'm pleased to inform you that your manuscript has been deemed suitable for publication in PLOS ONE. Congratulations! Your manuscript is now with our production department. 

Kind regards, 

on behalf of

Prof. Massimiliano Galdiero 

Academic Editor

PLOS ONE